# Expressive Gaussian Human Avatars
# from Monocular RGB Video

**Hezhen Hu**[1]    **Zhiwen Fan**[1]    **Tianhao Wu**[2]    **Yihan Xi**[1]    **Seoyoung Lee**[1]
**Georgios Pavlakos**[1]    **Zhangyang Wang**[1]
[1] University of Texas at Austin    [2] University of Cambridge

## Abstract

Nuanced expressiveness, especially through detailed hand and facial expressions, is pivotal for enhancing the realism and vitality of digital human representations. In this work, we aim to learn expressive human avatars from a monocular RGB video; a setting that introduces new challenges in capturing and animating fine-grained details. To this end, we introduce EVA, a drivable human model that can recover fine details based on 3D Gaussians and an expressive parametric human model, SMPL-X. Focused on enhancing expressiveness, our work makes three key contributions. First, we highlight the importance of aligning the SMPL-X model with the video frames for effective avatar learning. Recognizing the limitations of current methods for estimating SMPL-X parameters from in-the-wild videos, we introduce a reconstruction module that significantly improves the image-model alignment. Second, we propose a context-aware adaptive density control strategy, which is adaptively adjusting the gradient thresholds to accommodate the varied granularity across body parts. Third, we develop a feedback mechanism that predicts per-pixel confidence to better guide the optimization of 3D Gaussians. Extensive experiments on two benchmarks demonstrate the superiority of our approach both quantitatively and qualitatively, especially on the fine-grained hand and facial details. We make our code available at the project website: https://evahuman.github.io.

## 1 Introduction

High-quality digital avatar modeling has a wide range of applications, including AR/VR, movie production, sign language, and more. For digital human representation, capturing nuanced expressions is essential for enhancing fidelity and vitality. This is particularly evident in the detailed portrayal of hands and facial expressions, which add emotional depth and interactive expression capabilities to human avatars. In this work, we investigate expressiveness when building human avatars from monocular video. The task involves taking as input a monocular human video and learning an animated human avatar which enables multiple capabilities, such as novel view and pose synthesis.

Reconstructing expressive human avatars is challenging, particularly due to the subtle and complex movements of the hands and face. Compared to the body, hands and faces occupy smaller spatial areas and have distinct characteristics. For example, the hand has many degrees of freedom, intricate textures, and frequent self-occlusions. To achieve accurate avatar modeling, it is crucial to capture these fine textures from RGB video and ensure effective animation.

Current studies [49, 31, 15, 22, 12] mainly focus on learning human avatar on the body region and have made remarkable progress. Early works [49, 31, 15] mainly utilize NeRF as an implicit representation but usually have the drawback of low training/inference speed. Recently, more and more works build on top of 3D Gaussian Splatting [19] for its effectiveness and efficiency, which could further speed up rendering to over 100fps. GART [22] utilizes a mixture of moving 3D

38th Conference on Neural Information Processing Systems (NeurIPS 2024).

**SOTA Method**                                                                          **EVA**

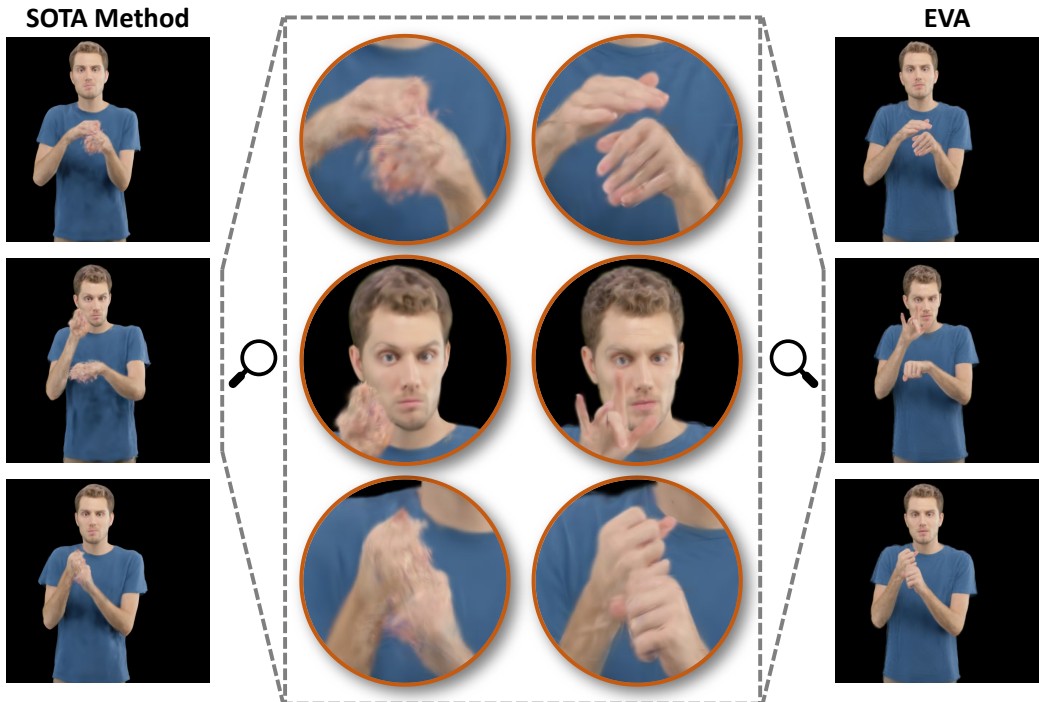

Figure 1: **Learning expressive Gaussian human avatars from RGB video.** Given a monocular video as input, our proposed approach, EVA, learns an expressive 3D Gaussian avatar. The results outperform the SOTA method [22] on novel pose synthesis, especially for the hand and facial details.

Gaussians to approximate human geometry and appearance and enhances fine details with learnable forward skinning and latent bones. GauHuman [12] proposes a new density control strategy, *e.g.* split and clone with KL divergence and a new merge operation. However, these methods do not consider the fine-grained details of hand and face, which cannot meet the requirement of expressiveness.

In this work, we introduce EVA, a drivable human model that can represent expressive details using 3D Gaussians and a human parametric model. Given a monocular RGB video, we extract the pose and mask information corresponding to each frame, which allows us to map each frame's human observation to a canonical space. Once the avatar is constructed, it can be animated using linear blend skinning given a new pose, followed by rendering to the 2D human image. To tackle new challenges introduced by expressiveness, we start by solving the misalignment issue between the SMPL-X model and the RGB frames via a reconstruction module. By employing a fitting-based optimization, this module can produce more reliable 3D SMPL-X reconstruction, providing a more robust foundation for digital avatar modeling.

Considering the granularity differences across different body parts, we propose a context-aware adaptive density control strategy for 3D Gaussian optimization. It leverages attributes specific to different parts and historical gradient information, to adaptively control Gaussian density. Furthermore, to improve the Gaussian optimization, we design a feedback mechanism, which adaptively predicts confidence scores based on the rendering result, thereby ensuring the supervision signal effectively transferring to the corresponding Gaussian. For evaluation on expressiveness, we build comparison baselines from related body avatar methods with a few modifications and collect a new benchmark called UPB containing in-the-wild upper body videos. The image quality is evaluated with multiple metrics, separately on the full body, as well as the hand and face regions. Our approach, as shown in Figure 1, largely outperforms previous state of the art, with more fidelity in fine-grained details.

Our contributions are summarized as follows:

- We introduce EVA, an approach that can build expressive human avatars based on 3D Gaussians, given as input a monocular RGB video. Through extensive experiments on two datasets, we demonstrate the superiority of EVA, particularly on the hand and facial details.

- To enhance expressiveness, we propose a context-aware adaptive density control strategy to accommodate the granularity differences across human parts, followed by a feedback mechanism to better guide the 3D Gaussian optimization.
- To handle challenging in-the-wild videos, we propose a reconstruction module, which critically improves the SMPL-X alignment to the RGB frames compared to off-the-shelf methods. We demonstrate the importance of this improvement in recovering accurate avatars.

## 2 Related Work

### 2.1 Human Avatar Modeling from Monocular RGB Video

The problem of building a human avatar from a monocular RGB video is challenging due to the dynamic nature of the capture and the partial observations for each frame. Early works [31, 2, 37, 38, 3, 15, 49, 46, 39, 42, 6, 21, 41] mainly resort to the combination of implicit neural representations (*e.g.,* NeRF [27]) and parametric models to represent a human avatar with high fidelity and flexibility. To address the slow computation in implicit models, previous work has proposed various techniques to reduce training [10, 14] or inference time [33, 25, 5]. With the introduction of 3D Gaussian Splatting [19] which can achieve both fast and high-quality rendering, an increasing number of works [22, 24, 11, 12, 20, 23, 32, 13, 18] utilize it as their base representation, jointly with parametric human models like SMPL [26]. More specifically, GaussianAvatar [11] enhances 3D Guassians via two key components for final photorealistic quality, *i.e.,* dynamic properties and joint optimization of motion and appearance. GauHuman [12] leverages human prior and KL divergence to propose a new density control strategy. In this work, we systematically explore expressiveness while building human avatars from a monocular RGB video and tackle the new challenges brought by expressiveness.

### 2.2 Expressive Human Representations

Expressiveness plays a vital role in non-verbal communication, which, besides the body, particularly involves the hand and face regions [44, 48, 51, 8, 9, 28, 16, 7, 35]. To represent expressive humans, various parametric models have been proposed, including SMPL-X [28], Adam [17], GHUM [43], and more. These models usually have predefined topology and can provide a compact mapping from the low-dimensional embedding to the 3D mesh. However, the predefined topology limits their capability of depicting fine-grained texture. Recent works build human avatars via neural 3D representations while leveraging the parametric model to provide human shape priors and animation signals. For example, X-Avatar [35] combines NeRF and SMPL-X to build the avatar. It further proposes part-aware sampling and initialization strategies to ensure efficient learning from high-quality 3D scans or RGB-D data. GVA [24] integrates the SMPL-X model to improve the rendering quality. AvatarRex [50] proposes a compositional avatar representation to separately model body, hand and faces from multi-view RGB video data. Different from them, we aim to relax the input requirements, by building expressive human avatars from a real-world monocular RGB video.

## 3 Technical Approach

In this section, we first introduce the preliminary knowledge on the SMPL-X human model [28] and 3D Gaussian Splatting [19] and present the general framework design on articulated human modeling. Then, we elaborate on the key technical contributions of our approach.

### 3.1 Preliminaries

**SMPL-X body model.** SMPL-X [28] is a parametric human body model, which extends the SMPL body model [26] by modeling both hand articulation and facial expressions. SMPL-X can be defined as a mapping function $M(\boldsymbol{\theta}, \boldsymbol{\beta}, \boldsymbol{\psi}) : \mathbb{R}^{|\boldsymbol{\theta}|} \times \mathbb{R}^{|\boldsymbol{\beta}|} \times \mathbb{R}^{|\boldsymbol{\psi}|} \to \mathbb{R}^{3N}$, where $\boldsymbol{\theta}, \boldsymbol{\beta}, \boldsymbol{\psi}$ are the parameters for pose, shape, and facial expression, respectively. The function of SMPL-X is formulated as follows:

$$\mathbf{M}(\boldsymbol{\beta}, \boldsymbol{\theta}, \boldsymbol{\psi}) = W(\mathbf{T}(\boldsymbol{\beta}, \boldsymbol{\theta}, \boldsymbol{\psi}), J(\boldsymbol{\beta}), \boldsymbol{\theta}, \mathbf{W}), \quad (1)$$

$$\mathbf{T}(\boldsymbol{\beta}, \boldsymbol{\theta}, \boldsymbol{\psi}) = \bar{\mathbf{T}} + B_S(\boldsymbol{\beta}) + B_E(\boldsymbol{\psi}) + B_P(\boldsymbol{\theta}), \quad (2)$$

where $B_P(\cdot)$, $B_S(\cdot)$ and $B_E(\cdot)$ denote pose, shape, and expression blend functions, respectively, and $\mathbf{W}$ is a set of blend weights. The pose, expression and shape corrective blend shapes, *i.e.,* $B_P(\boldsymbol{\theta})$,

$B_E(\boldsymbol{\psi})$ and $B_S(\boldsymbol{\beta})$, add corrective vertex displacements to the template human mesh $\bar{\mathbf{T}}$. After that, linear blend skinning $W(\cdot)$ is applied to rotate the vertices in the template mesh around the joints $J(\boldsymbol{\beta})$, smoothed by the blend weights $\mathbf{W}$. This generates the final human mesh.

**3D Gaussian Splatting.** Methods based on NeRF [27] model the scene with an implicit representation and render novel views using volume rendering. In contract, 3D Gaussian Splatting [19] (3DGS) models a 3D scene with a set of discrete 3D Gaussians and performs rendering through a tile-based rasterization operation, which can reach real-time rendering speeds. Specifically, each Gaussian is defined with its central position $p$, 3D covariance matrix $\Sigma$ as follows:

$$G(x) = exp(-\frac{1}{2}(x - p)^T \Sigma^{-1}(x - p)), \tag{3}$$

where $x$ is an arbitrary position in the 3D scene. The covariance matrix $\Sigma$ is decomposed into two learnable components to make the optimization easier, $\Sigma = RSS^T R^T$, where $R$ and $S$ denote the rotation matrix and scaling vector, respectively. During rendering, each 3D Gaussian $G(x)$ is first transformed to a 2D Gaussian $G'(x)$ on the image plane. Then, a tile-based rasterizer is designed to efficiently sort the 2D Gaussians and employ $\alpha$-blending:

$$C(r) = \sum_{i \in N} c_i \sigma_i \prod_{j=1}^{i-1}(1 - \sigma_j), \quad \sigma_i = \alpha_i G'(r), \tag{4}$$

where $r$ is the queried pixel position and N denotes the number of sorted 2D Gaussians associated with the queried pixel. $c_i$ and $\alpha_i$ denote the color and opacity of the $i$-th Gaussian, which are modeled by Spherical harmonics.

**Articulated 3D human modeling.** Our formulation takes inspiration from works employing 3D Gaussians for modeling articulated objects [12, 22]. We optimize 3D Gaussians in a canonical space, which corresponds to a human in a rest pose [15]. The Gaussians are transformed from the canonical space to the frame space via linear blend skinning (LBS):

$$\mathbf{p}^f = \mathbf{G}(\mathbf{J}^f, \boldsymbol{\theta}^f)\mathbf{p}^c + \mathbf{b}(\mathbf{J}^f, \boldsymbol{\theta}^f, \boldsymbol{\beta}^f), \tag{5}$$

$$\boldsymbol{\Sigma}^f = \mathbf{G}(\mathbf{J}^f, \boldsymbol{\theta}^f)\boldsymbol{\Sigma}^c \mathbf{G}^{\mathbf{T}}(\mathbf{J}^f, \boldsymbol{\theta}^f), \tag{6}$$

where $\mathbf{p}^f, \boldsymbol{\Sigma}^f, \mathbf{p}^c, \boldsymbol{\Sigma}^c$ are the Gaussian mean and covariance in frame space and canonical space, respectively. $\mathbf{G}(\mathbf{J}^f, \boldsymbol{\theta}^f) = \sum_{k=1}^{K} w_k \mathbf{G}_k(\mathbf{J}^f, \boldsymbol{\theta}^f), \mathbf{b}(\mathbf{J}^f, \boldsymbol{\theta}^f, \boldsymbol{\beta}^f) = \sum_{k=1}^{K} w_k \mathbf{b}_k(\mathbf{J}^f, \boldsymbol{\theta}^f, \boldsymbol{\beta}^f)$ are the rotation and translation, respectively, with respect to the $K$ joints, and $\mathbf{G}_k(\mathbf{J}^f, \boldsymbol{\theta}^f), \mathbf{b}_k(\mathbf{J}^f, \boldsymbol{\theta}^f, \boldsymbol{\beta}^f)$ are the rotation and translation, respectively, with respect to the $k$-th joint. $w_k$ is the LBS weight.

To perform Linear Blend Skinning, we need two important components, *i.e.,* the LBS weights and the input pose parameters. Learning LBS weights $w_k$ from scratch would be inefficient and can lead to a local optimum in the early stage of the training. Therefore, for each Gaussian, we start with the LBS weight of the nearest SMPL-X vertex and use an MLP $f_{\Theta_w}$ to predict an LBS weight offset $w'_k$ using the positionally encoded [27] Gaussian centers $\gamma(\boldsymbol{p}^c)$. $w_k$ is therefore defined as:

$$w_k = \frac{e^{\log(w_k^{\text{SMPL-X}} + \epsilon) + w'_k}}{\sum_{j=1}^{K} e^{\log(w_j^{\text{SMPL-X}} + \epsilon) + w'_j}}, \tag{7}$$

$$w'_k = f_{\Theta_w}(\gamma(\boldsymbol{p}^c)[k]), \tag{8}$$

where $w_k^{\text{SMPL-X}}$ is the LBS weight of the nearest SMPL-X vertex. We set $\epsilon = 10^{-8}$.

Starting from the input pose $\boldsymbol{\theta}^{\text{SMPL-X}}$, we further fine-tune it via an MLP-based network $f_{\Theta_\theta}$, which is jointly optimized during the 3D Gaussian optimization process. The actual poses $\boldsymbol{\theta}$ used for optimization and rendering are therefore obtained as follows:

$$\boldsymbol{\theta} = \boldsymbol{\theta}^{\text{SMPL-X}} \otimes f_{\Theta_\theta}(\boldsymbol{\theta}^{\text{SMPL-X}}), \tag{9}$$

where $\otimes$ represents the vector pointwise product.

### 3.2 SMPL-X Alignment for Real-World Videos

A key requirement for learning accurate human avatars is to initialize the optimization process with a reliable SMPL-X estimate. However, this can be challenging, particularly in real-world cases,

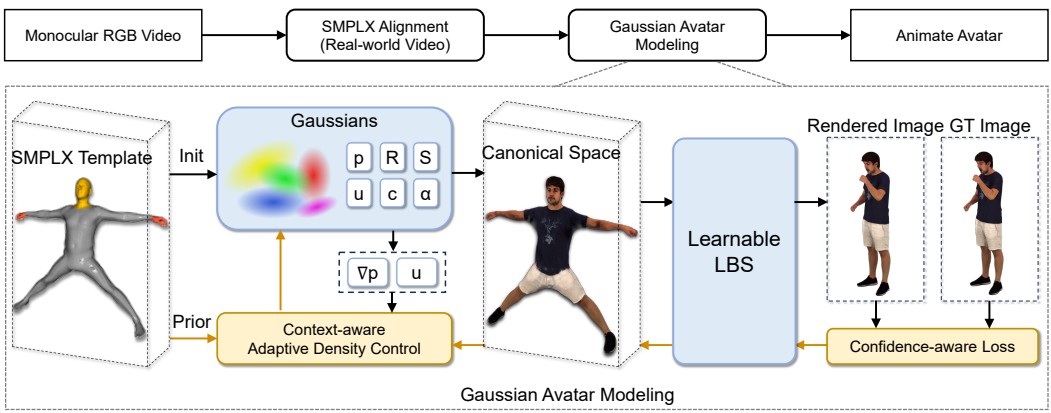

Figure 2: **Overview of our proposed EVA method**. Given a monocular RGB video, first we estimate a SMPL-X mesh that aligns well to the video frames using a reconstruction module. Then, EVA utilizes 3D Gaussian Splatting for avatar modeling, while inheriting the human shape prior from the SMPL-X model. To improve the optimization and the quality of the avatar, we propose context-aware adaptive density control and a confidence-aware loss.

given the limitations of current off-the-shelf methods [29] for SMPL-X fitting. This motivates us to propose a robust fitting procedure that leverages multiple sources, including the initial estimation of camera parameter, SMPL-X parameters, 2D keypoints and 3D hand parameters from off-the-shelf tools [30, 45, 1]. Given these initial estimates, the SMPL-X fitting procedure minimizes the following objective:

$$\mathcal{L}(\boldsymbol{\theta}, \boldsymbol{\beta}, \boldsymbol{\psi}) = \mathcal{L}_{2D} + \lambda_{bp}\mathcal{L}_{bp} + \lambda_{hp}\mathcal{L}_{hp}. \tag{10}$$

The 2D keypoint term, $\mathcal{L}_{2D}$, encourages the projection of the 3D keypoints of the avatar to align with the detected 2D keypoints. It is formulated as follows:

$$\mathcal{L}_{2D} = \sum_{i \in J} \gamma_i \omega_i \psi(\Pi_K(R_{\boldsymbol{\theta}}(J(\boldsymbol{\beta}))_i) - J_i^{2D}), \tag{11}$$

where $\Pi_K(\cdot)$ represents the camera projection under the given camera intrinsic parameter $K$. $\psi(\cdot)$ is the robust Geman-McClure error function [36] which helps prevent the disturbance from noisy 2D keypoint detections. $R_{\boldsymbol{\theta}}(\cdot)$ denotes the function which rotates the joints $J(\boldsymbol{\beta})$ given the pose $\boldsymbol{\theta}$. $J^{2D}$ is the detected 2D keypoints from [45]. $\gamma$ and $\omega$ represent the predefined weighting parameters and the detection confidences respectively.

Our prior terms contain two components which focus on the coarse-grained body and fine-grained hand, respectively. For the body part, we utilize VPoser [28] to filter infeasible body poses. VPoser provides a compact mapping from a low-dimensional embedding $\eta$ to the rotation matrices of the body pose $\theta_b$. To better optimize the low-dimensional embedding $\eta$, the estimated 3D body joints $J_b^{3D}$ [1] is treated as guidance, together with the added regularization term. The body prior loss term, $\mathcal{L}_{bp}$, is formulated as follows:

$$\mathcal{L}_{bp} = \psi(R_{\boldsymbol{\theta}}(J_b(\boldsymbol{\beta})) - J_b^{3D}) + ||\eta||^2. \tag{12}$$

Similary, for the hands, we utilize the 3D hand joints $J_h^{3D}$ estimated from [30] to provide a better hand spatial relationship. This is formulated as follows:

$$\mathcal{L}_{hp} = \psi_z(R_{\boldsymbol{\theta}}(J_h(\boldsymbol{\beta})) - J_h^{3D}), \tag{13}$$

where the index $h$ represents the joints corresponding to the hand. $\psi_z(\cdot)$ indicates that the robust error function only considers the coordinates along the z-axis coordinates for the error calculation.

### 3.3 Context-aware Adaptive Density Control

The core of the Gaussian Splatting optimization is adaptive density control, which generally contains two operations, *i.e.,* densification (split and clone) and pruning. The original strategy [19] selects a

fixed constant as the densification criteria. However, utilizing the fixed threshold does not leverage context information, leading to sub-optimal 3D Gaussian representations.

To this end, we propose a context-aware adaptive density control that leverages part attribute and history gradient information. Each Gaussian inherently possesses attributes associated with different body parts. These body parts vary in spatial size and characteristics. For instance, the hand is much smaller in size, compared to the body, exhibiting fine-grained textures and many degrees of freedom. These characteristics also lead to inherent differences in the Gaussian positional gradient changes across different body parts during the Gaussian optimization process. Furthermore, a continuous increase in the Gaussian positional gradient indicates that Gaussians need to be densified.

Specifically, we first initialize the 3D Gaussians with the vertices of the SMPL-X model. Since SMPL-X has a predefined topology, this initialization provides each Gaussian with the attribute information $U$ on which part it belongs, *e.g.* body, hands or face. After that, considering attributes and gradient history information, the densification threshold for the $i$-th Gaussian in a certain attribute $U$ is formulated as follows:

$$\epsilon_i = e + \frac{\lambda_t}{R} \Big( \sum_{k=t-R}^{t} \nabla_{i,k} - \sum_{k=t-2R}^{t-R} \nabla_{i,k} \Big), \tag{14}$$

where $e$ is a constant, $R$ represents the densification interval, and $\nabla_k$ denotes the position gradient of $i$-th Gaussian at the timestamp $k$. Note that $e$ and $\lambda_t$ have different values for different attributes. For the pruning strategy, we remove the points that are far away from the SMPL-X template vertices.

## 3.4 Objective Functions

The whole framework is optimized under the objective functions as follows:

$$\mathcal{L} = \mathcal{L}_c + \lambda_m \mathcal{L}_m + \lambda_s \mathcal{L}_{SSIM} + \lambda_l \mathcal{L}_{LPIPS}, \tag{15}$$

where $\lambda_m$, $\lambda_s$, and $\lambda_l$ are loss weighting terms. The mask loss $\mathcal{L}_m$ calculates the consistency between accumulated volume density and the estimated mask. The SSIM [40] loss $\mathcal{L}_{SSIM}$ is adopted to improve the structural similarity between rendered image and input image. The LPIPS [47] loss $\mathcal{L}_{LPIPS}$ focuses on the perceptual quality of the rendered image.

**Confidence-aware loss $\mathcal{L}_c$.** It is inevitable that any training video will introduce some form of noise (*e.g.*, misalignment, motion blur), which will also affect the avatar optimization procedure. To address some of the effects of the noise, we introduce a feedback module to decide which pixels should be taken into consideration with higher or lower weight during training. This module takes as input the rendered image $I_r$ and rendered depth $D_r$, and predicts a score for each pixel which represents the confidence value. More specifically:

$$C = \mu + exp(E(I_r, D_r)), \tag{16}$$

where $\mu$ is a constant. Then the confidence serves as an adaptive weighting factor on the per-pixel consistency. Eventually, we formulate our confidence-aware loss as follows:

$$\mathcal{L}_c = C \odot |I_r - I|_1. \tag{17}$$

## 4 Experiments

In this section, we first introduce our experimental setup, including datasets, implementation details and evaluation metrics. Then, we make comparisons with baseline methods both quantitatively and qualitatively. Finally, we perform ablation studies on the most important components of our approach.

### 4.1 Experimental Setup

**Datasets.** The experiments are conducted on two datasets, **XHumans** [35] and our collected **UPB** dataset. **XHumans** is a dataset captured in a controlled environment. It provides images with resolution of $1200 \times 800$, along with well-aligned SMPL-X meshes. There are 6 identities (3 male and 3 female) for evaluation. **UPB** consists of sign language videos from the web, which usually contain complicated hand gestures. The resolution of the videos is $1920 \times 1080$ and they do not contain any ground truth SMPL-X annotations. UPB includes 4 identities (2 male and 2 female).

Table 1: Comparison with three expressive avatar baselines, *i.e.*, GART + SMPLX, Splatting + SMPL-X and GauHuman + SMPLX, on the XHumans and UPB dataset. N-GS denotes the number of Gaussians. ↑ and ↓ represent the higher the better, and the lower the better, respectively.

| Method | N-GS | Full | | | Hand | | | Face | | |
|---|---|---|---|---|---|---|---|---|---|---|
| | | PSNR↑ | SSIM↑ | LPIPS↓ | PSNR↑ | SSIM↑ | LPIPS↓ | PSNR↑ | SSIM↑ | LPIPS↓ |
| *Controlled setting: XHumans dataset* | | | | | | | | | | |
| 3DGS [19] + SMPLX | 19,458 | 28.88 | 0.9609 | 44.93 | 25.28 | 0.9189 | 91.37 | 25.91 | 0.9087 | 101.04 |
| GART [22] + SMPLX | 89,571 | 27.73 | 0.9553 | 50.32 | 25.42 | 0.9151 | 99.53 | 25.86 | 0.9013 | 105.06 |
| Splatting [34] + SMPLX | 103,193 | 29.33 | 0.9606 | 44.39 | 26.19 | 0.9264 | 78.53 | 26.47 | 0.9103 | 92.51 |
| GauHuman [12] + SMPLX | 17,134 | 29.16 | 0.9623 | 41.16 | 25.69 | 0.9225 | 88.16 | 26.27 | 0.9124 | 93.35 |
| EVA | 19,993 | **29.67** | **0.9632** | **33.05** | **26.27** | **0.9279** | **72.95** | **26.56** | **0.9157** | **72.30** |
| *Real-world setting: UPB dataset* | | | | | | | | | | |
| 3DGS [19] + SMPLX | 21,008 | 25.31 | 0.9469 | 90.80 | 24.89 | 0.9425 | 66.19 | 24.57 | 0.9072 | 136.53 |
| GART [22] + SMPLX | 90,676 | 26.20 | 0.9511 | 78.90 | 25.25 | 0.9411 | 61.44 | 26.62 | 0.9253 | 93.28 |
| Splatting [34] + SMPLX | 257,811 | 25.13 | 0.9355 | 96.16 | 24.20 | 0.9298 | 70.91 | 24.48 | 0.8962 | 127.63 |
| GauHuman [12] + SMPLX | 12,372 | 25.17 | 0.9455 | 84.87 | 24.67 | 0.9418 | 67.61 | 24.33 | 0.9035 | 113.13 |
| EVA | 20,829 | **26.78** | **0.9519** | **65.07** | **27.00** | **0.9524** | **45.90** | **26.85** | **0.9298** | **65.90** |

**Implementation details.** Our framework is implemented on PyTorch and all experiments are performed on NVIDIA A5000. The hyperparameter $\lambda_m$, $\lambda_s$ and $\lambda_l$ are set to 0.1, 0.01 and 0.04, respectively. We use SGHM [4] to extract the human mask. The 3D Gaussian optimization lasts for 2,000 iterations with the densification performed between iteration 400 and 1,000. For other parameters, we follow the original settings of [19]. Our method can render a 1080p image with an inference speed of 361.02 fps on one RTX A5000. Since our method is reconstruction-based, our main requirement is that the areas of interest should be captured in the video. Note that we use the same input RGB frames for all methods to ensure a fair comparison. More details are presented in the Appendices.

**Evaluation metrics.** We evaluate the avatar quality through the rendered views following previous works [22, 12]. We adopt the widely-used Peak Signal-to-Noise Ratio (PSNR), Structural Similarity Index Measure (SSIM) [40], and Learned Perceptual Similarity (LPIPS) [47] to evaluate the rendered images. For a more fine-grained evaluation, we report the above metrics separately for the full body, for the hands region and for the face region.

## 4.2 Comparison with baselines

The baselines we use for comparison are adopted from related methods with a few modifications. 3DGS + SMPL-X is modified from 3DGS [19]. We add our articulated human modeling mechanism to make it fit the task requirement. GART and GauHuman are originally designed to model body-level avatars. They are animated via the SMPL parametric model, which lacks the capability of modeling hand articulation and facial expressions. Therefore, we replace the driven signal with the SMPL-X model and denote them as GART + SMPL-X and GauHuman + SMPL-X, respectively. Splatting selects the triangle mesh as the driving signal and we utilize SMPL-X mesh in Splatting + SMPL-X. Among these baselines, GauHuman is closer to our approach. Our differences are mainly reflected in the adaptive density control strategy, objective functions and the SMPL-X alignment for the real-world videos, which are also consistent with our main technical contributions.

As shown in Table 1, we conduct experiments on the XHumans and UPB datasets to evaluate the effectiveness of our method. XHumans is captured in a controlled setting with accurate SMPL-X ground truth, while UPB focuses on real-world video without pose information. Since it is hard to get accurate SMPL-X annotation for real videos, UPB is more challenging and more representative of learning a human avatar in the wild. We observe that our method achieves state-of-the-art performance on these two datasets. We note that we do not apply our designed alignment module in XHumans, since it already contains SMPL-X ground truth. Our method achieves 19.7%, 17.3% and 22.5% relative LPIPS gain on the full, hand and face regions, respectively. For the real-world UPB dataset, our performance gain is much larger, achieving over 25% relative LPIPS gain on the hand region. This indicates that EVA, unlike previous work, handles well the challenges of real-world videos. The qualitative comparison in Figure 3 also validates the effectiveness of our method.

| GT | 3DGS+ | GART+ | GauHuman+ | EVA |
|----|-------|-------|-----------|-----|

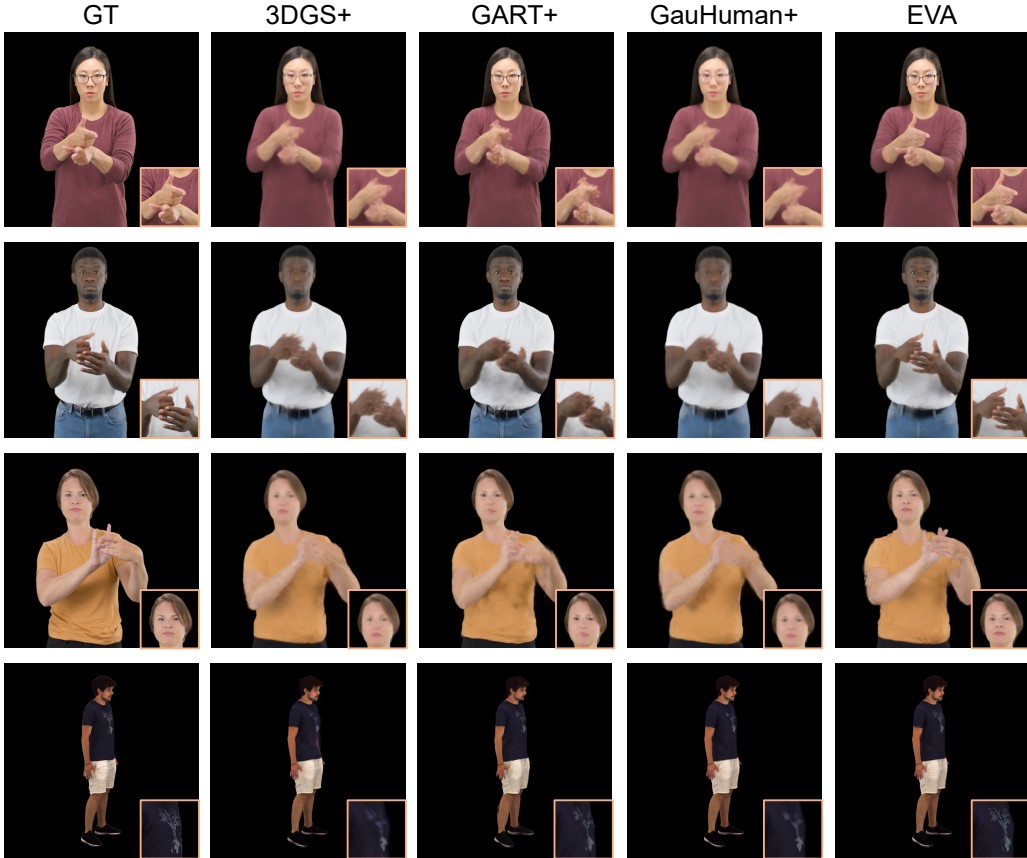

Figure 3: **Qualitative comparison with baselines.** We compare with 3DGS [19] + SMPL-X, GART [22] + SMPL-X, and GauHuman [12] + SMPL-X. First three rows are from UPB and last row is from the XHumans dataset. EVA exhibits the best visual quality. See the zoomed-in results in the box for comparison of the fine-grained details.

## 4.3 Ablation Studies

We perform ablation experiments to highlight the important components of EVA, *i.e.*, context-aware adaptive density control, confidence-aware loss and in-the-wild SMPL-X alignment module.

**Effectiveness of context-aware adaptive density control (CADC) and confidence-aware loss (CL).** These two components are designed for better optimization of *Gaussian Avatar Modeling*. As shown in Table 2, "w/o CADC" means that we replace it with the original density control [19], while "w/o CL" denotes removing the confidence weighting term in the RGB loss calculation. We observe that these two components both improve performance. More specifically, CADC improves performance for the metrics of all regions. Meanwhile, we observed that CL brings another benefit of a more compact representation (e.g., reducing the number of Gaussians from 21,038 to 19,993).

**Effectiveness of in-the-wild SMPL-X alignment.** We use our SMPL-X alignment module for real-world videos without accurate SMPL-X ground truth, so we conduct the corresponding ablation study on UPB dataset. As shown in Table 2, "w/o Align" means that we directly utilize the SMPL-X mesh extracted from current SOTA estimation method [1]. Notably, our proposed alignment module brings notable performance gains on all metrics of all regions. It also demonstrates the importance of the SMPL-X pose quality for final human avatar modeling. Without accurate SMPL-X alignment, the following Gaussian model will need to deal with inconsistent textures for a specific region across frames, leading to lower avatar quality. Furthermore, we visualize the SMPL-X alignments in Figure 4 and present qualitative comparison with state-of-the-art estimation methods in Figure 5, including fitting-based SMPLify-X and regression-based SMPLer-X. Visible improvements are clearly observed specifically for the hands.

Table 2: Effect of context-aware adaptive density control (CADC), confidence-aware loss (CL) and SMPL-X alignment. ↑ and ↓ represent "higher the better", and "lower the better", respectively.

| Method | Full | | | Hand | | | Face | | |
|---|---|---|---|---|---|---|---|---|---|
| | PSNR↑ | SSIM↑ | LPIPS↓ | PSNR↑ | SSIM↑ | LPIPS↓ | PSNR↑ | SSIM↑ | LPIPS↓ |
| *XHumans dataset* | | | | | | | | | |
| w/o CADC | 28.92 | 0.9611 | 35.14 | 25.23 | 0.9175 | 84.00 | 26.08 | 0.9080 | 80.76 |
| w/o CL | 29.63 | **0.9634** | 34.08 | 26.27 | 0.9279 | 74.02 | **26.58** | **0.9166** | 73.19 |
| EVA | **29.66** | 0.9632 | **33.05** | **26.27** | **0.9279** | **72.95** | 26.56 | 0.9156 | **72.30** |
| *UPB dataset* | | | | | | | | | |
| w/o Align | 25.02 | 0.9435 | 73.82 | 24.64 | 0.9396 | 63.64 | 24.27 | 0.9009 | 93.56 |
| w/o CADC | 26.53 | 0.9504 | 68.76 | 26.43 | 0.9494 | 50.59 | 26.45 | 0.9265 | 71.17 |
| w/o CL | 26.74 | 0.9507 | 67.90 | 26.74 | 0.9507 | 48.54 | 26.69 | 0.9293 | 69.60 |
| EVA | **26.72** | **0.9519** | **65.37** | **26.90** | **0.9523** | **46.11** | **26.75** | **0.9298** | **66.41** |

| GT | w/o Align | EVA | GT | w/o Align | EVA |
|---|---|---|---|---|---|

Figure 4: **Effect of our SMPL-X alignment module**. We can estimate a SMPL-X mesh that aligns well with the RGB frame, especially for the fine-grained hand regions.

## 4.4 More Discussion

In summary, we apply EVA on a variety of monocular RGB videos, including both controlled captures and videos collected from the Internet. The visual data contain large body motions (e.g., playing basketball, weight lifting, dancing), along with fine-grained motions (e.g. finger counting, using tools, sign language). EVA can handles well self-occlusions, which usually exist in every frame, especially for the hand areas. This can be attributed to our proposed SMPL-X fitting method which provides reliable correspondences from the pixels to the canonical space. Besides, we expect that EVA could also handle low lighting conditions (as long as the lighting is not changing significantly), since our SMPL-X fitting method should be robust to low lighting conditions. Given accurate SMPL-X estimates, we expect the avatar learning stage will perform well.

**Limitations.** The focus of this work is to capture expressive human avatars, by designing a pipeline that can take raw web videos as the input. Besides capturing the expressive details, other factors are challenging, yet worth investigating to further improve the quality of the avatar, *e.g.* non-rigid elements like clothes and hair. It is also worth extending our avatar reconstruction to more challenging input sources, *e.g.* occlusions or few-shot scenarios.

**Failure Cases.** We show some representative examples of our failure cases in Figure 6. For the fine-grained expressive areas, the failure cases happen when the video includes hand interactions. The failure is mainly caused by incorrect SMPL-X reconstruction (e.g., the estimated SMPL-X model that drives the reconstruction has implausible interpenetration between hands). Another failure case is the existence of floaters in the reconstruction which is a common issue of 3DGS modeling.

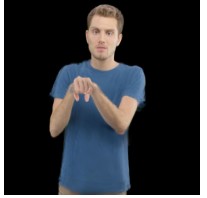 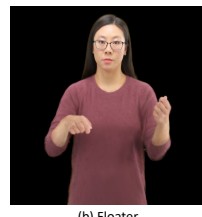

(a) False driving signal      (b) Floater

Figure 6: **Illustration of failure cases.**

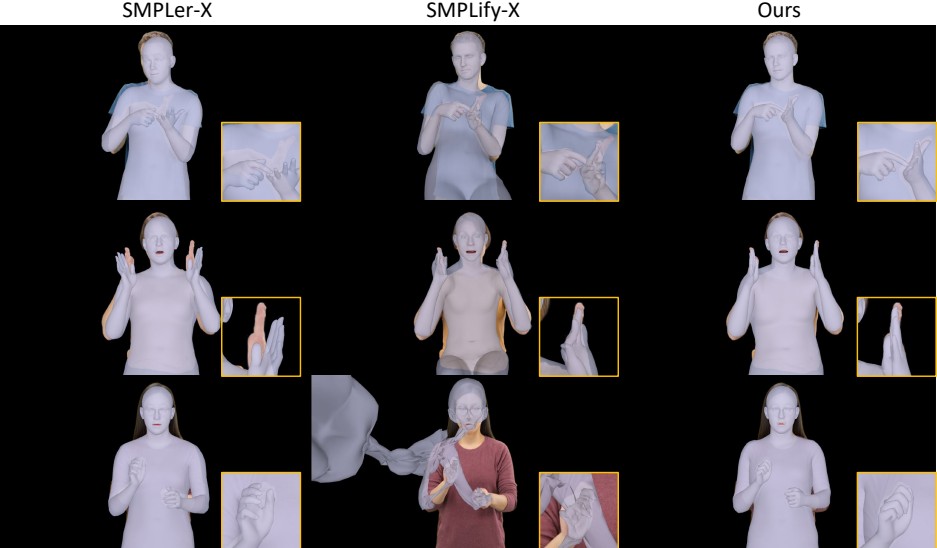

Figure 5: **Comparison among methods for SMPL-X alignment.** We compare our results with a regression-based method (SMPLer-X), and a fitting-based method (SMPLify-X).

**Broader Impact.** In this work, our proposed framework helps push the boundaries of what is possible with human avatar modeling, allowing for more expressive results than previously achievable. However, with these advancements come certain risks, particularly concerning the potential misuse of the technology for deceptive practices or harassment. By advancing SOTA, our work makes these potential risks more pressing, as the barrier to creating such deceptive or harmful content is lowered. This highlights the need for careful consideration of the ethical implications and the development of strategies to mitigate these risks.

## 5   Conclusion

In this work, we present EVA, a drivable expressive human avatar learned from a real-world monocular RGB video. EVA is built on 3D Gaussians, in coordination with the human prior introduced by SMPL-X. To deal with the challenges brought by expressiveness, we first utilize a reconstruction module to solve the misalignment between SMPL-X model and RGB frames. During the optimization of the 3D Gaussians, we propose context-aware adaptive density control, which leverages attribute and historical gradient information to accommodate the varied granularity across body parts. A feedback mechanism is jointly designed to further guide learning. Extensive experiments on two benchmarks demonstrate that our method outperforms baselines both quantitatively and qualitatively, especially on the fine-grained hand and facial details.

**Acknowledgments.** This project was supported by LUCI program under the Basic Research Office and partially supported by ARL grants W911NF20-2-0158 and W911NF-21-2-0104 under the cooperative A2I2 program. It was also in part supported by NSF AI Institute for Foundations of Machine Learning (IFML). GP has received a research gift from Google.

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

# Appendices

This technical appendices provides more details which are not included in the main paper due to space limitations. For more visualization results, please refer to the project webpage.

**Training/Testing split.** For each identity in XHumans dataset, one video is selected as the training split, where the other videos are marked as testing. During training, we utilize all of the frames (150 frames). We sample 20 frames for each testing video with the sampling rate of 5. For UPB dataset, we uniformly sample the frames with the interval as 1 to split the training and testing frames. The number of training and testing frames are both 140.

**Additional implementation details.** $\mu$ is set to 1. $\lambda_t$ is set as -9.0, -4.5 and -6.3 for body, hand, and face parts, respectively. We set $e$ for body, hand, and face parts as 2e-4, 1e-4 and 1.4e-4, respectively. The feedback module $E(\cdot)$ consists of two 2D convolutional networks. For SMPL-X alignment, we utilize the L-BFGS optimizer with the Wolfe line search. The optimization contains three stages with different loss weighting factors. The first stage is designed to initialize the human body embedding to the initial estimation from the method [1]. Then the second stage mainly aims to get better spatial hand relationship leveraging the prediction from the method [30]. The third stage performs fine-tuning with more emphasize on the fine-grained hand and face regions.

**Comparison baselines.** The baselines we use for comparison are adopted from related methods with a few modifications. 3DGS + SMPL-X is modified from 3DGS [19]. Since the original 3DGS does not have the capability of animation, we add the same articulated human modeling in Section 3.1 as ours. For fair comparison, it also utilizes the same optimization schedule as ours, which lasts 2,000 iterations with the densification performed between 400 and 1,000 iterations. GART and GauHuman are originally designed to model body-level avatars. They are animated via the SMPL parametric model, which lacks the capability of modeling hand articulation and facial expressions. Therefore, we replace the driven signal with the SMPL-X model and denote them as GART + SMPL-X and GauHuman + SMPL-X, respectively. Since Splatting is embedded on a triangle mesh, we utilize the SMPL-X human mesh as its driving signal and denote it as Splatting + SMPL-X. We do not modify the optimization schedules of these three methods.

**Future works.** We outline the potential future works as follows,

- Modeling capability on non-rigid elements, such as loose cloth (dress). Modeling cloth on the avatar has been a challenging topic, even separately studied in several works. A potential solution could be parameterizing the cloth deformation or adding more prior on cloth type, so as to provide more driving signals.
- Generalizable human avatar from monocular RGB video. Current methods in this topic mostly need per-subject optimization, which needs to be re-trained to any new given subjects. It is worth exploring if we could get an avatar from a monocular RGB video of any given subject with a single feed-forward pass.
- Robustness. It is worth exploring if we could build the human avatar well with more limited source inputs, e.g. a few images with even mutual occlusion, a single image, etc.

