# OpenReview forum: "Expressive Gaussian Human Avatars from Monocular RGB Video"
_NeurIPS.cc/2024/Conference — NeurIPS 2024 poster_

### Official Review · Reviewer_ZKQ6 · 2024-07-10

**Soundness:** 3
**Presentation:** 3
**Contribution:** 3
**Rating:** 5
**Confidence:** 4

**Summary:**

The paper focuses on improving the expressiveness of digital human avatars, particularly through detailed hand and facial expressions, learned from monocular RGB video. The main contributions are:
- SMPL-X Alignment improves the alignment of the SMPL-X model with RGB frames and aids in accurately recovering avatars.
- Context-Aware Adaptive Density Control Strategy adjusts gradient thresholds to handle the varied granularity across different body parts, enhancing the expressiveness of the avatars.
- Feedback Mechanism that predicts per-pixel confidence to better guide the learning and optimization of 3D Gaussians.

Overall, the paper presents a comprehensive framework that enhances the realism and expressiveness of digital human representations, validated by substantial quantitative and qualitative experiments.

**Strengths:**

The paper is written in a clear and understandable manner. Comprehensive ablation studies and clear visualizations help to demonstrate the effectiveness of different components of the method. The paper also compares the new method with previous methods and shows that it achieves better performance. It is commendable that the authors added SMPL-X to existing methods for fair and thorough comparisons.

**Weaknesses:**

- One of the contributions of using and optimizing SMPL-X for expressive avatar is not novel. This has been done in previous works [1].
- The paper lacks a comparison to recent methods such as Splatting Avatar [2]. Including this comparison in Table 1 can help to distinguish the improvement made by the proposed approach.
- There is an inconsistent ablation study across the datasets. Table 2 should illustrate how the alignment affects the baseline method by comparing scenarios with and without CADC, CL, and alignment. Similarly, Table 3 should also show the method with CADC and CL included.
- PSNR, SSIM, and LPIPS should be calculated from four novel views to ensure the robustness of the method across different perspectives. Additionally, animation and novel pose metrics can be included to display differences between methods.
- From Table 2, it seems that CL does not have a significant impact. This should be addressed or clarified to understand its role in the framework. It would be good to provide more qualitative visualization to demonstrate its effectiveness.


[1] (Liu et al., 2024) GVA: Reconstructing Vivid 3D Gaussian Avatars from Monocular Videos.

[2] (Shao et al., 2024) SplattingAvatar: Realistic Real-Time Human Avatars with Mesh-Embedded Gaussian Splatting.

**Questions:**

- **Project Page for Visualizations:** Is there a project page available that shows visualizations of the rendered method, such as a 360-degree rotating video of the generated person and animation to better illustrate the quality? Specifically, could visualizations zoom into detailed areas such as the hands and face?
- Is e and λt in Equation 14 learned or fixed? If learned, it would be advantageous to list the values to demonstrate that the learned parameters effectively handle different resolution details for different body parts.
- Discussing failure cases and outlining potential future work would provide a more comprehensive view of the method's limitations and areas for improvement.
- Information on the optimization time compared to previous methods and the resource requirements for running the proposed method would be valuable for assessing its efficiency and scalability.

**Limitations:**

While the use of CADC is interesting, the other two contributions are of limited novelty. SMPL-X usage and alignment have been incorporated by previous works. The improvement of CL is not significant. The addition of project page for visualizations would be helpful to demonstrate the improvements made by the methods. In addition, more quantitative results should be added to Tables 1, 2 and 3.

---

> ### Author Rebuttal · Authors · 2024-08-07
>
> **Q1: One of the contributions of using and optimizing SMPL-X for expressive avatar is not novel. This has been done in previous works GVA.**
>
> A1: GVA is a concurrent work and has not been accepted. Compared with GVA, our fitting design is quite different, since ours explicitly focuses on the fine-grained areas (especially on hand) to meet the requirement of expressiveness. Considering GVA is not open-sourced, we are not able to compare with it. However, we have visualized that our fitting method outperforms the current SOTA SMPL-X estimation method in both Figure 4 (main paper) and Figure 1 (rebuttal pdf).
>
> **Q2: More comparison with recent methods such as SplattingAvatar.**
>
> A2: Thanks for this suggestion. The results of SplattingAvatar are shown in A1 of General Response. We will add these experiment results in Table 1.
>
>
> **Q3: Clarification on inconsistent ablation study across the datasets.**
>
> A3: Table 2 demonstrates the ablation results on the XHumans dataset. For XHumans, we directly utilize its provided accurate SMPLX annotation. That is to say, we do not need the SMPL-X fitting involved. For Table3, we further update the ablation results.
>
> | **Method** | **PSNR (Full)** | **SSIM (Full)** | **LPIPS (Full)** | **PSNR (Hand)** | **SSIM (Hand)** | **LPIPS (Hand)** | **PSNR (Face)** | **SSIM (Face)** | **LPIPS (Face)** |
> |-------------------|--------------------|--------------------|-----------------------|--------------------|--------------------|-----------------------|--------------------|--------------------|-----------------------|
> | w/o Align  | 25.02 | 0.9435 | 73.82 | 24.64 | 0.9396 | 63.64 | 24.27 | 0.9009 | 93.56
> | w/o CADC   | 26.53 | 0.9504 | 68.76 | 26.43 | 0.9494 | 50.59 | 26.45 | 0.9265 | 71.17
> | w/o CL     | 26.74 | 0.9507 | 67.90 | 26.74 | 0.9507 | 48.54 | 26.69 | 0.9293 | 69.60
> | EVA        | 26.72 | 0.9519 | 65.37 | 26.90 | 0.9523 | 46.11 | 26.75 | 0.9298 | 66.41
>
>
>
> **Q4: PSNR, SSIM, and LPIPS should be calculated from four novel views to ensure the robustness of the method across different perspectives. Additionally, animation and novel pose metrics can be included to display differences between methods.**
>
> A4: We calculate PSNR, SSIM and LPIPS from different viewpoints. To our best knowledge, we cannot find any animation and novel pose metrics reported in relevant works (GART, GauHuman, SplattingAvatar, etc.)
>
> **Q5: Clarification on CL role in the framework.**
>
> A5: We observed that CL helped with perceptually improving the results, which is also reflected in the consistent performance improvement on the LPIPS metrics. Moreover, another benefit of CL is that it led to a more compact representation (e.g., reducing the number of Gaussians from 21,038 to 19,993). We further present some qualitative results (w/o and w confidence-aware loss) in Figure 3 of the rebuttal pdf.
>
> **Q6: Project page for visualization including a 360-degree rotating video, animation, and zooming.**
>
> A6: Due to the NeurIPS policy ("all the texts you post (rebuttal, discussion and PDF) should not contain any links to external pages"), we are unable to provide the external link to our built project page. In fact, we have included 360-degree rotating video (Supp video) and animation (Supp video) and zooming results (both Supp video and figures in main paper). We promise to release the project page afterward.
>
> **Q7: Is e and λt in Equation 14 learned or fixed?**
>
> A7: Both e and λt are fixed.
>
> **Q8: Discussion on failure cases and outlining potential future work**.
>
> A8: Thanks for pointing out this issue. For the fine-grained expressive areas, the failure cases happen when hand interaction exists, which is mainly caused by the false driven SMPLX signal (e.g. the driven SMPLX has implausible interpenetration between hands). From the holistic view, some 'floaters' may exist sometimes, where general 3DGS modeling also exists.
>
> We outline the potential future works as follows,
>
> 1) Modeling capability on non-rigid elements, such as loose cloth (dress). Modeling cloth on the avatar has been a challenging topic, even separately studied in several works. A potential solution could be parameterizing the cloth deformation or adding more prior on cloth type, so as to provide more driving signals.
>
> 2) Generalizable human avatar from monocular RGB video. Current methods in this topic mostly need per-subject optimization, which needs to be re-trained to any new given subjects. It is worth exploring if we could get an avatar from a monocular RGB video of any given subject with a single feed-forward pass.
>
> 3) Robustness. It is worth exploring if we could build the human avatar well with more limited source inputs, e.g. a few images with even mutual occlusion, a single image, etc.
>
> **Q9: Information on the optimization time compared to previous methods and the resource requirements.**
>
> A9: For the optimization time, we generally divide it into two stages. Take a real-world 1080p video as an example, the first stage (data preprocessing) needs 10.2 minutes, while the second stage (avatar modeling) needs 7.5 minutes on RTX A5000. Compared with the avatar modeling stage of EVA, 3DGS+SMPLX needs 7.5min, GART+SMPLX needs 12.9 min, GauHuman+SMPLX needs 4.5min, and SplattingAvatar needs 45.2 min. The resource requirement could be further lowered down to consumer GPU cards like 3090, but at the expense of speed.

---

> ### Comment · Reviewer_ZKQ6 · 2024-08-13
>
> Thank you for the detailed responses. I also recommend including the fixed values of e and λt​ in the paper, along with a justification for how these values were determined. Given the overall quality of this submission, I will recommend the acceptance of this paper. Please include the additional experimental results in your revision.

---

> > ### Author Response · Authors · 2024-08-14
> >
> > Thank you for recognizing our efforts in addressing your concerns. We are pleased that our responses have been helpful and will incorporate the additional experimental results and responses into the main paper, as per your suggestions.
> >
> > We are always open to further discussion that could help enhance our paper. If there are no further concerns, we would greatly appreciate it if you could kindly consider raising the rating.
> >
> > Thank you again for your valuable input.

---

### Official Review · Reviewer_g7vJ · 2024-07-10

**Soundness:** 3
**Presentation:** 3
**Contribution:** 4
**Rating:** 6
**Confidence:** 4

**Summary:**

The paper introduces a 3DGS-based human avatar generation method from a monocular RGB video. Specifically, the authors first optimize the SMPL-X model to better align with the RGB frames. Then, they propose an adaptive adjustment method for different body parts and use per-pixel confidence to guide 3DGS. It is reported that the proposed method achieves state-of-the-art performance. Visual results show fine-grained hand and facial details.

**Strengths:**

1. Generating the human upper body videos is a critical task. It is more expressive and more challenging than facial generation. This method overall demonstrates fine-grained generation details.
2. The existing SMPL-X model has alignment issues when estimating the human body. The optimized SMPL-X model in this method significantly improves the alignment between SMPL-X and the RGB frames.
3. The paper is overall easy to follow, and the experimental results demonstrate the method's superiority in numerical metrics.

**Weaknesses:**

1. In the paper, the confidence scores for CL are all learned from the rendered results. How can the effectiveness of this confidence be ensured? In Table 2, the face area without CL also shows better results.
2. [11] also demonstrates photorealistic 3D human body generation, and the authors need to include more discussions with it.
3. The paper has many writing issues, such as a too-brief related work section, inconsistent formula definitions (e.g., $\Pi_K$ in Eq. 11), and some missing punctuation in formulas.

**Questions:**

1. What are the specific requirements for videos in terms of the method, such as the length of the video and the range of camera angles?
2. How does the method perform when the magnitude of the action being driven is out-of-distribution?
3. What is the inference speed of the method?

**Limitations:**

This paper has involved discussions about limitations and broader impact.

---

> ### Author Rebuttal · Authors · 2024-08-07
>
> **Q1: How can the effectiveness of this confidence be ensured?**
>
> A1: The predictor could leverage the information contained in rendered RGB and depth to adaptively learn the confidence in an end-to-end data-driven manner. The experimental results have validated the effectiveness of CL and we also present some visualization in Figure 3 of the rebuttal pdf. Another benefit of CL is to reduce the number of Gaussians from 21,038 to 19,993.
>
> **Q2: GaussianAvatar also demonstrates photorealistic 3D human body generation, and the authors need to include more discussions with it.**
>
> A2: GaussianAvatar is one of the pioneering animatable 3D Gaussian models which is learned from a monocular RGB video. Its representation is further enhanced via two key components for final photorealistic quality. Dynamic properties are designed to support pose-dependent appearance modeling, while joint optimization of motion and appearance helps tackle inaccurate motion estimation. We will add this discussion in related work.
>
> **Q3: Suggestion on revising some writing issues, such as a too-brief related work section, inconsistent formula definitions (e.g., in Eq. 11), and some missing punctuation in formulas.**
>
> A3: Thanks for pointing out these issues. We will discuss more works in the related work section, e.g. GaussianAvatar, fix the inconsistent formula definitions, and add the missing commas in Eq. 7, 8, 9, 10 and 13. We will incorporate these modifications in the revised manuscript.
>
> **Q4: Specific requirements for videos in terms of the method.**
>
> A4: Since our method is reconstruction-based, our main requirement is that the areas of interest should be captured in the video. For example, for the full-body X_Human dataset, the monocular RGB video includes observations around the person of interest. On the other hand, for the upper-body UPB dataset, since the area of interest for sign language is only on the upper body, we only need to capture the corresponding appearance of the person. We note that we keep the input RGB frames of all comparison methods consistent for fair comparison.
>
> **Q5: How does the method perform when the magnitude of the action being driven is out-of-distribution?**
>
> A5: It is hard to quantify the magnitude of out-of-distribution actions in a principal way (although if the reviewer would like to indicate a specific metric/evaluation, we would be happy to explore it). With that being said, on the UPB benchmark for instance, our evaluation is conducted on novel poses that are not used during training, so our evaluation indicates the performance on the novel pose setting. Moreover, qualitatively, we demonstrate the results of our avatar driven by an in-the-wild SMPL-X sequence with unseen poses and unseen identities. By inspecting the results, the expressive details are quite faithful. However, if the motion used at test time is completely different, it is likely we will observe artifacts in the reconstruction.
>
> **Q6: Inference speed of the method.**
>
> A6: Our method could render the 1080p image with the inference speed of 361.02 fps on one RTX A5000.

---

### Official Review · Reviewer_fFxo · 2024-07-12

**Soundness:** 3
**Presentation:** 4
**Contribution:** 3
**Rating:** 6
**Confidence:** 5

**Summary:**

The paper presents EVA, a model that can recover expressive human avatars from monocular RGB videos. The primary focus is enhancing fine-grained hand and facial expressions using 3D Gaussian Splatting and the SMPL-X model. EVA introduces three key contributions: a plug-and-play module for better SMPL-X alignment, a context-aware adaptive density control strategy, and a feedback mechanism for optimizing 3D Gaussian learning. Extensive experiments on two benchmarks demonstrate EVA's superiority in capturing detailed expressiveness compared to previous methods.

**Strengths:**

The paper introduces three significant innovations: a plug-and-play alignment module, a context-aware adaptive density control strategy, and a feedback mechanism for 3D Gaussian optimization.
2. Extensive quantitative and qualitative evaluations on two benchmarks show EVA's effectiveness in capturing fine-grained details, particularly in hand and facial expressions.
3. The proposed method addresses the challenging task of creating expressive avatars from monocular RGB video, which has practical applications in VR/AR, movie production, and video games.
4. The paper comprehensively explains the technical approach, including SMPL-X alignment, Gaussian optimization, and adaptive density control.

**Weaknesses:**

1. The proposed method's complexity might pose implementation challenges for those not well-versed in the field. Simplifying some aspects or providing more intuitive explanations could help.
2. While the method shows superior performance on the provided datasets, the generalizability to other types of monocular videos or environments is not thoroughly discussed.
3. Although the paper compares EVA with previous SOTA methods, it would be beneficial to include more diverse baselines or variants to demonstrate the robustness of the proposed approach.

**Questions:**

1. How does EVA perform with videos that have significant occlusions or low lighting conditions?
2. Are there any specific preprocessing steps required for the input monocular RGB videos?
3. Can the proposed method be extended to capture other expressive elements, such as cloth or hair movements?

**Limitations:**

1. The performance heavily relies on the quality of the datasets used. Real-world applications might present challenges not covered by the current benchmarks.
2. The method requires significant computational power, which might limit its applicability in real-time scenarios or on devices with limited resources.
3. While the method focuses on hand and facial details, other expressive elements like body language, clothing, and hair dynamics are not addressed.

---

> ### Author Rebuttal · Authors · 2024-08-07
>
> **Q1: Suggestion on simplifying some aspects or providing more intuitive explanations.**
>
> A1: Thanks for this suggestion. We commit to releasing our code upon acceptance. Our work makes a step towards building an expressive avatar from the real-world video. In short, our method aims to answer two key questions below,
>
> 1) How to generate a well-aligned SMPLX mesh to provide accurate correspondence for expressive avatar modeling? We propose a fitting-based alignment method explicitly focusing on fine-grained details (e.g. hand).
>
> 2) How to enhance the expressiveness during avatar modeling? We propose context-aware adaptive density control, in coordination with the feedback mechanism.
>
> **Q2: More discussion on the generalizability to other types of monocular videos or environments.**
>
> A2: Currently, our work has included various types of monocular RGB videos from controlled capturing setting to those collected from the Internet. The involved human poses contain large body motion (e.g., playing basketball, weight lifting, do kicks, dancing), along with fine-grained motions (e.g. finger counting, using tools, sign language). Among them, considering the complexity and practicability of sign language, we separately involve it to evaluate expressiveness and collect a new benchmark (UPB) from the Internet. We acknowledge that there are more types of videos and environments that could be further explored (such as when the people are occluded by other objects, changing light conditions) and will add this discussion as promising future work.
>
> **Q3: Suggestion on including more diverse baselines or variants to demonstrate the robustness of the proposed approach.**
>
> A3: Thanks for your suggestion! We have used the Splatting Avatar method from CVPR 24 and evaluated it on the datasets we used. The results are shown in A1 of General Response. We will add these experiment results in Table 1.
>
>
> **Q4: How does EVA perform with videos that have significant occlusions or low lighting conditions?**
>
> A4: EVA can deal well with self-occlusions, which usually exist in each frame, especially on the hand areas. This can be attributed to our proposed SMPL-X fitting method which could provide reliable correspondence to map the pixels to the canonical space. Both qualitative and quantitative results demonstrate the effectiveness of our method. Besides, we expect that EVA could also handle low lighting conditions (as long as the lighting is not changing significantly), since our SMPL-X fitting method should be robust to low lighting conditions and with good SMPL-X estimates, we expect the Gaussian Splatting stage to perform well.
>
> **Q5: Are there any specific preprocessing steps required for the input monocular RGB videos?**
>
> A5: Since our method is reconstruction-based, our main requirement is that the areas of interest should be captured in the video. For example, for the full-body X_Human dataset, the monocular RGB video includes observations around the person of interest. On the other hand, for the upper-body UPB dataset, since the area of interest for sign language is only on the upper body, we only need to capture the corresponding appearance of the person. We note that we keep the input RGB frames of all comparison methods consistent for fair comparison.
>
> **Q6: Can the proposed method be extended to capture other expressive elements, such as cloth or hair movements?**
>
> A6: We believe our work could be extended to capture other expressive elements. One possible solution is to embed the driving signal with more modeling capabilities considering the other expressive elements that we need to incorporate. In this way, our method could benefit from explicit signals to perform modeling. However, this is beyond the scope of our current work.
>
>
> **Q7: The performance heavily relies on the quality of the datasets used. Real-world applications might present challenges not covered by the current benchmarks.**
>
> A7: The high-quality dataset is needed to ensure sufficient resolution on fine-grained expressive areas for training and evaluation. With the popularization of low-cost consumer cameras, high-quality data (e.g. 1080p) becomes much easier to capture, even via personal phones. With that being said, our proposed SMPLX fitting simplifies and robustifies the whole avatar generation pipeline, such that it works with sufficient robustness in Internet videos (e.g. see the UPB videos that come from YouTube).
>
> **Q8: The method requires significant computational power, which might limit its applicability in real-time scenarios or on devices with limited resources.**
>
> A8: We only need one GPU for both training and testing. Once our model is trained, it is efficient to run, which could render the 1080p image at 361.02 fps on one RTX A5000.

---

> > ### Comment · Reviewer_fFxo · 2024-08-13
> > **Thanks for the responses**
> >
> > Thanks for the detailed responses. Given the overall quality of this submission, I will recommend the acceptance of this paper.

---

> > > ### Author Response · Authors · 2024-08-14
> > >
> > > Thank you for recognizing our efforts in addressing your concerns. We feel very grateful that you will recommend our work for acceptance. We will revise the manuscript as you suggested.

---

### Official Review · Reviewer_4SGk · 2024-07-14

**Soundness:** 3
**Presentation:** 2
**Contribution:** 2
**Rating:** 5
**Confidence:** 3

**Summary:**

This paper proposes a solution to generate expressive human avatars from monocular RGB videos. The main focus is to improve the expressiveness. To this end, a few ideas such as combining 3d Gaussian splatting and SMPL-X (SMPL+parametric hands), minimizing 2d reprojection error,  finegrained density control, etc.

There are a few similar papers in citation. To better understand the novelty of this paper, here's a high level comparison:
+ SMPLer-X [1]: Representation is SMPL-X only. Input is monocular RGB videos. Losses: SMPL-X keypoint losses.
+ GART[23]:  Representation is 3d Gaussians+SMPL. Input is monocular RGB videos.  Losses: photometric, perceptual.
+ 3DGS-Avatar [33]: Representation is 3d Gaussians+SMPL; Input is monocular RGB videos; Losses: photometric, perceptual, mask, etc;
+ This paper: Representation is 3d Gaussians+LBS+SMPL-X. Input is monocular RGB videos. Losses: SMPL-X keypoint losses, perceptual.

**Strengths:**

From the high level comparison, the paper does find a unique and timely combination of 3d Gaussians and SMPL-X, which is well-motivated by improving the expressiveness. Some of the ideas such as adaptive density control are therefore new as it leverages the fine-grained details of SMPL-X.

**Weaknesses:**

The concept of model and image alignment by minimizing reprojection losses is not new. For example, the keypoints losses (Sec. 3.2) are very common in literature [1, 30].
Similarly mask loss and perceptual loss can be found in [33].

The adaptive density control part seems novel, as it's initialized from body parts and leveraging the fine-grained topology of SMPL-X.

The confidence-aware loss seems new but only provides marginal improvement as in Table 2.

Lack of hyperparameter ablation. Not even in supplementary.

Minor issue: `video` is countable. Please explicitly use `videos` or `a video` in the paper. Actually it's not clear how many videos of one subject are used to train one avatar.

**Questions:**

What's 'E' in Eq (16)? It looks like the so-called feedback module. But what's the design? Is it a learning-based predictor?

Please also address the novelty questions in the weakness part.

**Limitations:**

Limitations are discussed. It'd be good to discuss the failure cases.

---

> ### Author Rebuttal · Authors · 2024-08-07
>
> **Q1: Clarification on reprojection losses.**
>
> A1: We want to clarify that the reprojection losses (keypoint, mask, perceptual loss) are not claimed as our contributions. Instead, we include them in our paper to describe the necessary components of our framework and make sure our paper is self-contained.
>
> **Q2: Clarification on confidence-aware losses.**
>
> A2: We observed that the Confidence-aware loss helped with perceptually improving the results, which is also reflected to the consistent performance improvement on the LPIPS metrics. Moreover, another benefit of using the Confidence-aware loss is that it led to a more compact representation (e.g., reducing the number of Gaussians from 21,038 to 19,993). We also present some qualitative results (w/o and w confidence-aware loss) in Figure 3 of the rebuttal pdf.
>
> **Q3: Ablation on hyperparameter.**
>
> A3: We observed that one of the most crucial hyperparameters was the value $\lambda_{t}$ in the context-aware adaptive density control. We perform an ablation on this hyperparameter.
>
> | **$\lambda_{t}$** | **PSNR (Full)** | **SSIM (Full)** | **LPIPS (Full)** | **PSNR (Hand)** | **SSIM (Hand)** | **LPIPS (Hand)** | **PSNR (Face)** | **SSIM (Face)** | **LPIPS (Face)** |
> |-------------------|--------------------|--------------------|-----------------------|--------------------|--------------------|-----------------------|--------------------|--------------------|-----------------------|
> | 0.0               | 28.92              | 0.9611             | 35.14                 | 25.23              | 0.9175             | 84.00                 | 26.08              | 0.9080             | 80.76                 |
> | -6.0              | 29.28              | 0.9619             | 33.71                 | 25.69              | 0.9221             | 77.81                 | 26.33              | 0.9114             | 74.21                 |
> | **-9.0**              | **29.66**          | **0.9632**             | **33.05**             | **26.27**          | **0.9279**         | **72.95**             | **26.56**              | **0.9156**             | **72.30**             |
> | -12.0             | 29.29              | 0.9619             | 33.85                 | 25.67              | 0.9219             | 78.75                 | 26.34              | 0.9113             | 74.29                 |
>
>
>
> **Q3: Typo on 'video'.**
>
> A3: Thanks for pointing out this issue. We will revise the usage of 'video' in the manuscript. We clarify that only one monocular RGB video of one subject is used to train the corresponding avatar.
>
> **Q4: What's 'E' in Eq (16)? It looks like the so-called feedback module. But what's the design? Is it a learning-based predictor?**
>
> A4: 'E' is the feedback module. It mainly consists of two convolutional layers and is a learning-based predictor. We show the pseudo code of this module below.
>
> ```
> class E_feedback(nn.Module):
>     def __init__(self, num_feat=4):
>         super(E_feedback, self).__init__()
>         self.conv1 = nn.Conv2d(num_feat, 3, 1, 1, 0)
>         self.conv2 = nn.Conv2d(num_feat + 3, 1, 1, 1, 0)
>         self.lrelu = nn.LeakyReLU(negative_slope=0.2, inplace=True)
>     def forward(self, x):
>         x1 = self.lrelu(self.conv1(x))
>         x2 = self.conv2(torch.cat((x, x1), 1))
> ```
>
> **Q5: Suggestion on discussing the failure cases.**
>
> A5: Thanks for pointing out this issue. For the fine-grained expressive areas, the failure cases happen when hand interaction exists, which is mainly caused by incorrect SMPLX reconstruction (e.g. the estimated SMPLX model that drives the reconstruction has implausible interpenetration between hands). Another failure case is the existence of floaters in the reconstruction which is a common issue of 3DGS modeling. We show some representative examples in Figure 2 of the rebuttal pdf.

---

> > ### Comment · Reviewer_4SGk · 2024-08-13
> >
> > Thanks for the clarification. Please include these in the revision.

---

> > > ### Author Response · Authors · 2024-08-14
> > >
> > > Thank you for recognizing our efforts in addressing your concerns. We are glad that our responses have been helpful and will incorporate these responses into the revised manuscript as you suggested. We are always open to discussions that could further improve our paper. If there are no additional concerns, we would greatly appreciate it if you could kindly consider raising the rating.
> > >
> > > Thank you again for your valuable comments.

---

### Official Review · Reviewer_1sDc · 2024-07-18

**Soundness:** 3
**Presentation:** 3
**Contribution:** 2
**Rating:** 6
**Confidence:** 4

**Summary:**

- Given an RGB human video, the paper focuses on photorealistic reconstruction of the human in 3D using SMPL-X and Gaussian rendering.
- The main idea is to align the expressive SMPL-X with the video evidence followed by Gaussian Avatar modeling with better gaussian density control and loss.
- Technical contributions: Context-aware adaptive density control, confidence-aware loss, SMPL-X alignment.
- Baselines: 3DGS, GART, GauHuman.
- Evaluation are done on two datasets, XHuman and UPB. Metrics: PSNR, SSIM, LPIPS.
- The results show that the proposed method EVA consistently outperforms the baselines.

**Strengths:**

- The paper is well written, organized and easy to follow.
- The focus problem is of great importance for the field of human digitization.
- The ablative studies are informative, giving us insights into the importance of the technical components.

**Weaknesses:**

- Weak technical contributions: In comparison to GauHuman, CVPR 2024, the proposed method EVA makes three changes. 1) Context-aware adaptive density (CADC), 2) Confidence-aware loss (CL), 3) SMPL-X alignment. CADC is a nice idea for human modeling, and is inspired by the original 3DGS paper. However, it is a slight change to the originally proposed heuristic. CL is frequently used in the field of pose-estimation and 3D modeling (eg. DUST3R, CVPR 2024). SMPL-X alignment is very simply mesh fitting to 2D keypoints and is well explored in SMPLify-X, CVPR 2019 and its related works. Overall, the main technical ideas proposed lack technical novelty.

- XHuman over ZJU_MoCap for evaluations: Is there a reason why metrics were reported on XHuman and not ZJU-MoCap dataset? The missing metrics on the popular ZJU-MoCap benchmark makes it hard to fairly compare EVA with existing. I am guessing this because ZJU-MoCap does not have SMPL-X annotations. In this context, can we use the same setup as UPB and still evaluate using predicted SMPL-X parameters?

**Questions:**

Given the lack of technical novelty, it is important to establish the empirical usefulness of the proposed method.
In this context, I am asking for a fair evaluation with prior methods (mostly using SMPL) on popular benchmarks.

**Limitations:**

Yes.

---

> ### Author Rebuttal · Authors · 2024-08-07
>
> **Q1: Clarification on technical contributions**.
>
> A1: Our proposed three technical contributions are well-motivated and well-formulated to meet the requirement of expressiveness in human avatar modeling, which are validated by extensive experiments and have been acknowledged by Reviewer 4SGk ('the paper introduces three significant innovations').
>
> Our proposed fitting aims at mitigating the misalignment issues, especially on the fine-grained hands, which designs effective loss terms leveraging multiple pseudo ground truths (meshes and 2D keypoints predicted by off-the-shelf methods). As shown in Figure 1 of the rebuttal pdf, our fitting results are clearly better than both SOTA regression-based method SMPLer-X and optimization-based fitting (SMPLify-X). Its effectiveness has also been acknowledged by Reviewer g7vJ ('significantly improves the alignment between SMPL-X and the RGB frames').
>
> **Q2: Reasons for not reporting results on ZJU-Mocap.**
>
> A2: The main reason for not choosing ZJU-Mocap is that it is not suitable for evaluating expressiveness. It has limited diversity on hand gestures/facial expressions (only clenched fist or open palm), and insufficient hand resolutions following the practice of previous methods.
> Meanwhile, we also prioritize the experiment on ZJU_Mocap and have spent over 24 hours trying to solve it.
> 1) The first option (proposed by the reviewer) is that we use the same setup as UPB and still evaluate using predicted SMPL-X parameters, the input SMPLX quality will be worse than the SMPL counterpart given in ZJU-Mocap, which will lead to an unfair comparison.
> 2) Another option is that we try to convert the given SMPL annotation to the form of SMPL-X. However, we still find there remains the projection discrepancy which will also lead to an unfair comparison.
>
> We have conducted the experiments on two benchmarks to perform a fair comparison with previous methods, which is acknowledged by Reviewer ZKQ6 ('It is commendable that the authors added SMPL-X to existing methods for fair and thorough comparisons').
>
>
> **Q3: Clarification on the empirical usefulness of the proposed method.**
>
> A3: The main empirical usefulness is that our method could build the expressive avatar from monocular RGB video while getting rid of time-consuming SMPLX annotation. From the application perspective, it could be utilized in sign language production. We have demonstrated this application by conducting experiments on the UPB benchmark (videos collected from the Web).

---

> > ### Comment · Reviewer_1sDc · 2024-08-13
> >
> > Thank you for the rebuttal. Appreciate your efforts on incorporating ZJU-Mocap. Overall, my concerns are addressed.
> > I am recommending the work for acceptance.

---

> > > ### Author Response · Authors · 2024-08-14
> > >
> > > We are glad that we could address your concerns and feel very grateful that you raised the score and recommended our work for acceptance. We will incorporate our responses in the revised manuscript.

---

### Author Rebuttal · Authors · 2024-08-07

**General Response**

We sincerely appreciate the reviewers for their insightful and constructive comments. We are encouraged by their positive comments and the recognition of the merits of our work. More specifically, the reviewers have appreciated the great importance and challenges introduced by expressiveness in human avatar modeling (R-1sDc, R-fFxo, and R-g7vJ). Moreover, they have highlighted the novelty (R-fFxo, R-4SGk, and R-g7vJ) and strong performance (R-fFxo, R-1sDc, R-g7vJ, and R-ZKQ6) of the proposed EVA framework for capturing expressiveness. Finally, the reviewers have positively commented on our comprehensive experiments (R-fFxo, R-1sDc and R-ZKQ6), the clear visualizations (R-fFxo, R-ZKQ6, and R-g7vJ), and the well-written and easy-to-follow paper (R-1sDc, R-fFxo, R-g7vJ and R-ZKQ6).

In the following, we first address the common concern, and then the concerns of each reviewer. We will revise the manuscript accordingly.

**Q1: Suggestion on including more comparison methods.**

A1: We have used the Splatting Avatar method from CVPR 24 and evaluated it on the datasets we used. The results are shown below. Our approach outperforms Splatting Avatar.
| **Method** | **Dataset** | **N-GS**   | **PSNR (Full)** | **SSIM (Full)** | **LPIPS (Full)** | **PSNR (Hand)** | **SSIM (Hand)** | **LPIPS (Hand)** | **PSNR (Face)** | **SSIM (Face)** | **LPIPS (Face)** |
|------------|-------------|------------|-----------------|-----------------|------------------|-----------------|-----------------|------------------|-----------------|-----------------|------------------|
| SplattingAvatar   |  UPB     | 257,811  | 25.13           | 0.9355          | 96.16            | 24.20           | 0.9298          | 70.91            | 24.48           | 0.8962          | 127.63           |
| EVA |  UPB     | 20,829  | 26.78           | 0.9519          | 65.07            | 27.00           | 0.9524          | 45.90            | 26.85           | 0.9298         | 65.90           |
| SplattingAvatar   | XHumans | 103,193  | 29.33           | 0.9606          | 44.39            | 26.19           | 0.9264          | 78.53            | 26.47           | 0.9103          | 92.51            |
| EVA | XHumans | 19,993 | 29.67 | 0.9632 | 33.05 | 26.27 | 0.9279 | 72.95 | 26.56 | 0.9157 | 72.30 |

We will add the discussion on this paper in related work and the experiment results in Table 1.

---

### Comment · Area_Chair_1xL7 · 2024-08-13
**Response to Authors**

Dear Reviewer,

The discussion period with the authors is coming to a close in less than 24 hours, at 11:59 PM AOE Aug 13, 2024. There is still time to review the authors' response to your questions and engage in a discussion with them. It would great if you could acknowledge having read the authors' response and update your rating if it changes your mind.

Best,
AC

---

### Decision · Program_Chairs · 2024-09-25

**Decision:**

Accept (poster)

**Comment:**

This paper proposes a new method called EVA for expressive 3D human avatar creation from monocular videos. To solve this task, the work proposes three innovations: improved SMPL-X estimation, context aware densification of Gaussians and context aware pixel-level feedback, which all improve quality. The method is evaluated on the X-Humans dataset, but more importantly on a new challenging UPB dataset of in-the-wild videos curated from the Internet. The proposed method shows clear improvements over the existing state of the art methods. Reviewers, appreciated the importance of the challenging problem addresses, and the novelty and good results of the proposed method. They initially raised concerns about the lack of comparisons to some baselines and the lack of ablations. The reviewers' concerns were addressed during the rebuttal phase and all reviewers were in agreement in recommending acceptance of this work. The AC concurs and recommends acceptance. The authors should include the changes that they have promised in the rebuttal into the final manuscript.